# The spatio-temporal evolution of multiple myeloma from baseline to relapse-refractory states

Leo Rasche [1,2,3], Carolina Schinke [1], Francesco Maura [4], Michael A. Bauer [1], Cody Ashby [1], Shayu Deshpande[1], Alexandra M. Poos[5], Maurizio Zangari[1], Sharmilan Thanendrarajan[1], Faith E. Davies[6], Brian A. Walker [7], Bart Barlogie[1], Ola Landgren [4], Gareth J. Morgan [6], Frits van Rhee [1] & Niels Weinhold[1,5] ✉

Deciphering Multiple Myeloma evolution in the whole bone marrow is key to inform curative strategies. Here, we perform spatial-longitudinal whole-exome sequencing, including 140 samples collected from 24 Multiple Myeloma patients during up to 14 years. Applying imaging-guided sampling we observe three evolutionary patterns, including relapse driven by a single-cell expansion, competing/co-existing sub-clones, and unique sub-clones at distinct locations. While we do not find the unique relapse sub-clone in the baseline focal lesion(s), we show a close phylogenetic relationship between baseline focal lesions and relapse disease, highlighting focal lesions as hotspots of tumor evolution. In patients with ≥3 focal lesions on positron-emission-tomography at diagnosis, relapse is driven by multiple distinct sub-clones, whereas in other patients, a single-cell expansion is typically seen ($p < 0.01$). Notably, we observe resistant sub-clones that can be hidden over years, suggesting that a prerequisite for curative therapies would be to overcome not only tumor heterogeneity but also dormancy.

Deciphering the evolutionary trajectories shaping the clonal architecture of Multiple Myeloma (MM) from disease initiation to relapsed-refractory disease is the key to inform novel curative treatment strategies. Recent longitudinal studies in MM have identified a number of genetic aberrations, which are associated with increased clonal fitness and can lead to relapse. These aberrations frequently arise through branching pathways, especially in patients relapsing from deep responses[1–7]. However, the origin of relapse subclones, as well as the extent of clonal diversity in treatment-resistant disease, remain largely unknown.

Crucial hints to the origin of MM cells, which are the source of relapse, comes from functional imaging studies, which have identified nodular plasma cell (PC) accumulations in the remission bone marrow (BM) of some patients with undetectable minimal-residual disease (MRD) at the iliac crest[8–10]. Genomic sequencing of these nodules, which are called focal lesions, showed site-specific enrichment of aberrations associated with relapse[11]. While these observations support focal lesions as being the origin of resistant disease and providing one possible explanation for their negative prognostic impact[12–14], the evolutionary pathways undergone by focal lesion subclones after treatment have not been defined.

Clonal diversity provides the fuel for drug resistance and treatment failure. Since current MM therapies induce deep responses, thereby potentially eradicating at least some subclones, it would be

[1]Myeloma Center, University of Arkansas for Medical Sciences, Little Rock, AR, USA. [2]Department of Internal Medicine 2, University Hospital of Würzburg, Würzburg, Germany. [3]Mildred Scheel Early Career Center (MSNZ), University Hospital of Würzburg, Würzburg, Germany. [4]Myeloma Program, Sylvester Comprehensive Cancer Center, University of Miami, Miami, FL, USA. [5]Department of Internal Medicine V, University Hospital of Heidelberg, Heidelberg, Germany. [6]Perlmutter Cancer Center, New York University Langone Health, New York, NY, USA. [7]Division of Hematology Oncology, Indiana University, Indianapolis, IN, USA. ✉e-mail: niels.weinhold@med.uni-heidelberg.de

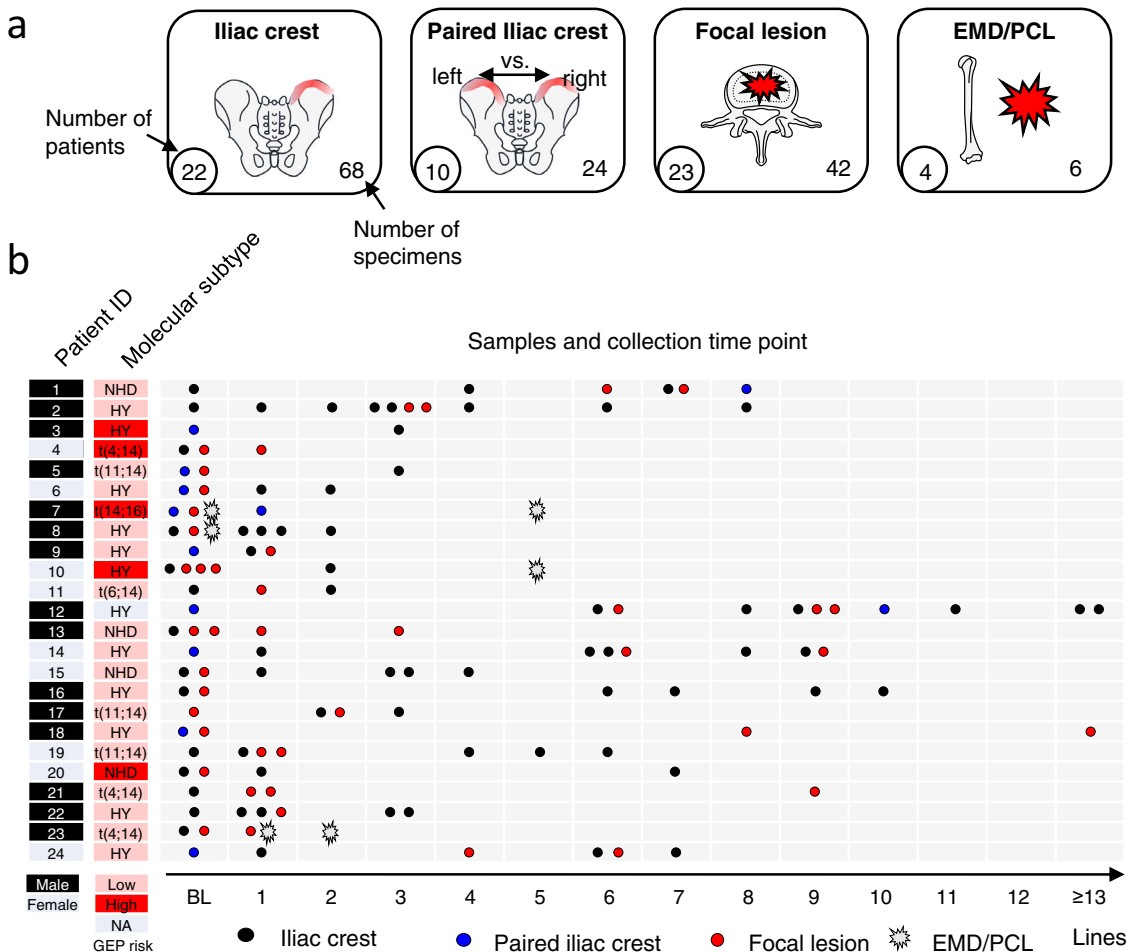

**Fig. 1 | Patient characteristics, sample origin, and collection time points.** In **a** the origin (red marking) and the number of investigated samples are depicted. In **b** gender, molecular subtype, risk according to the GEP70[19], as well as the type of samples, and the time points of sample collection are shown. EMD extramedullary disease, PCL plasma cell leukemia, NHD non-hyperdiploid karyotype, HY hyperdiploid, NA not available, Lines number of treatment lines (for a detailed description of treatment see Suppl. Data 1). Source data are provided as a Source Data file.

predicted that diversity decreases over time. According to the observations by ref. 15, a number of relapse subclones compete for access to the BM niche. In contrast to this model, the recent description of melphalan-induced mutational signatures in tumor cells derived from multiple sites suggests that a single clonal initiating cell can drive MM relapse[16,17]. Consequently, the exact model of how clonal diversity is generated in MM during therapy and its impact on resistance remains controversial. This deficiency could be remedied by the addition of spatial and temporal data on the clonal structure.

In this work, we analyze longitudinal multi-region whole-exome sequencing data together with matched clinical annotations, including a total of 140 tumor samples from 24 MM patients (Fig. 1 and Suppl. Data 1). By tracking resistant clones in time and space we dissect the sophisticated spatiotemporal evolutionary pathways that are active in MM during treatment and demonstrate the power of combining functional imaging and tumor sequencing to delineate a more complete picture of subclonal evolution in MM.

## Results

### Subclonal spatial heterogeneity is linked to the composition of focal lesions

Recently, we characterized focal lesions as hotspots of regional tumor evolution in MM[11] but the extent of spatial genomic heterogeneity outside of the focal lesions in the diffuse BM infiltrate remains unknown. From a clinical point of view, this information is important, since, in daily clinical practice, the left or right iliac crest are usually randomly chosen for diagnostic BM biopsies. To fill this gap in knowledge, we analyzed paired left and right iliac crest samples, which had been collected from 9 patients within a few days of each other at baseline (Fig. 1a, b).

We found a high degree of homogeneity in these paired samples with a median of 98% (range 96–100%) of mutations being shared, and in addition, there were only minor subclonal differences between the two sites (Fig. 2a, b). The homogeneous subclonal mix at the iliac crest is an observation that is reassuring, supporting that a second independent random biopsy usually does not provide additional information. In contrast, we observed frequent major unshared mutations in paired baseline iliac crest and focal lesion samples with a significantly lower proportion of shared mutations (median 79% (62–96%), $p = 0.001$, Fig. 2a, c). The picture seen at the iliac crest barely changed during therapy. Although we found shared-diff mutations (detectable in both samples but >3x CCF difference, see methods), there were no unshared major mutations and a median of 97% (93–98%) of mutations were shared between paired iliac crest samples (Fig. 2b). In paired iliac crest/ focal lesion or focal lesion/focal lesion pairs, however, there was significant genomic heterogeneity at a similar level to that seen at baseline (Fig. 2c).

Compared to these paired samples, even larger differences in genomic profiles were seen, when we compared the first to the last

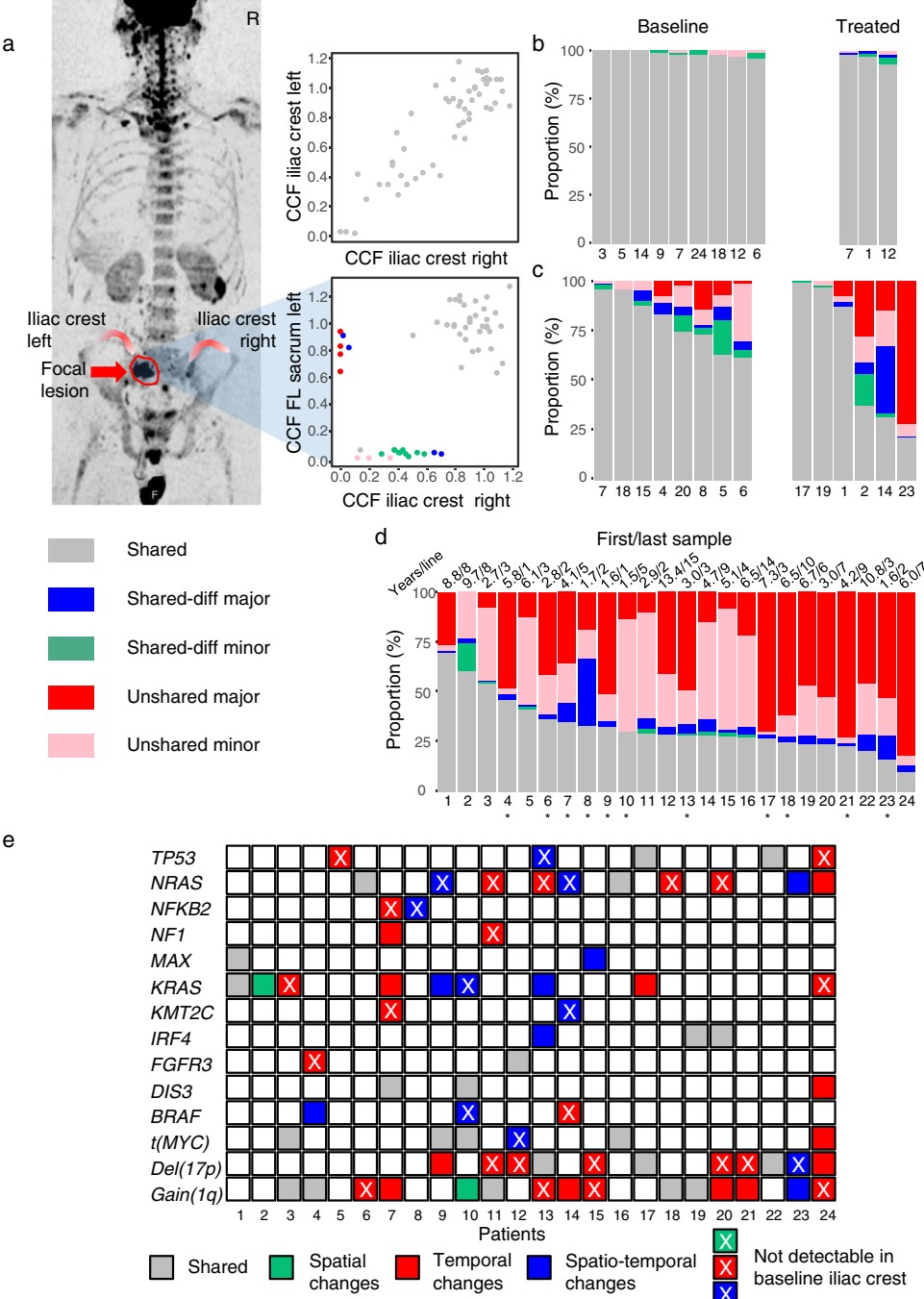

**Fig. 2 | Spatial heterogeneity in time and space.** We compared mutational profiles of paired samples collected from different bone marrow sites at baseline and during treatment, and discriminated between randomly collected samples from the iliac crest and CT-guided specimens from focal lesions. In **a** the functional imaging and the corresponding cancer clonal fraction (CCF) plots for detected mutations for patient #5 with paired samples from the right and left iliac crest as well as a sample from a left sacrum focal lesion are shown. In **b**, **c** the proportion of shared and heterogeneous mutations in paired left/right iliac crest and paired iliac crest/focal lesion samples collected at baseline and within the same treatment line is presented, respectively. For this comparison, paired samples from patients #10, #13, #16, and #23 (baseline) as well as #9, #12, #22, and #24 (treated) were not considered, as they had already been included in our previous analysis of spatial heterogeneity[11]. In **d** the mutational profiles of the first and the last available sample per patient are compared. For patients, where an iliac crest sample was compared to a focal lesion or a peripheral blood sample, the ID is marked with a star. Mutations were called shared (gray color), if they were present with the same or similar cancer clonal fraction (CCF) in paired samples. For heterogeneous mutations we discriminated between subclonal (CCF <60%) and clonal (CCF ≥60%) mutations. Mutations with at least a threefold difference in CCF were classified as shared-differential (subclonal: green; clonal: blue). We called mutations unshared if they were detectable in only one of the paired samples (subclonal: pink; clonal: red). In **e**, the spatiotemporal changes affecting known myeloma driver aberrations are presented, including MYC translocations, gain(1q), del(17p), as well as myeloma driver genes[22], which showed non-silent mutations in at least two patients in our set. Gray indicates shared (clonal) events. Green, red, and blue denote spatial, temporal, and spatiotemporal differences/changes, respectively. Source data are provided as a Source Data file.

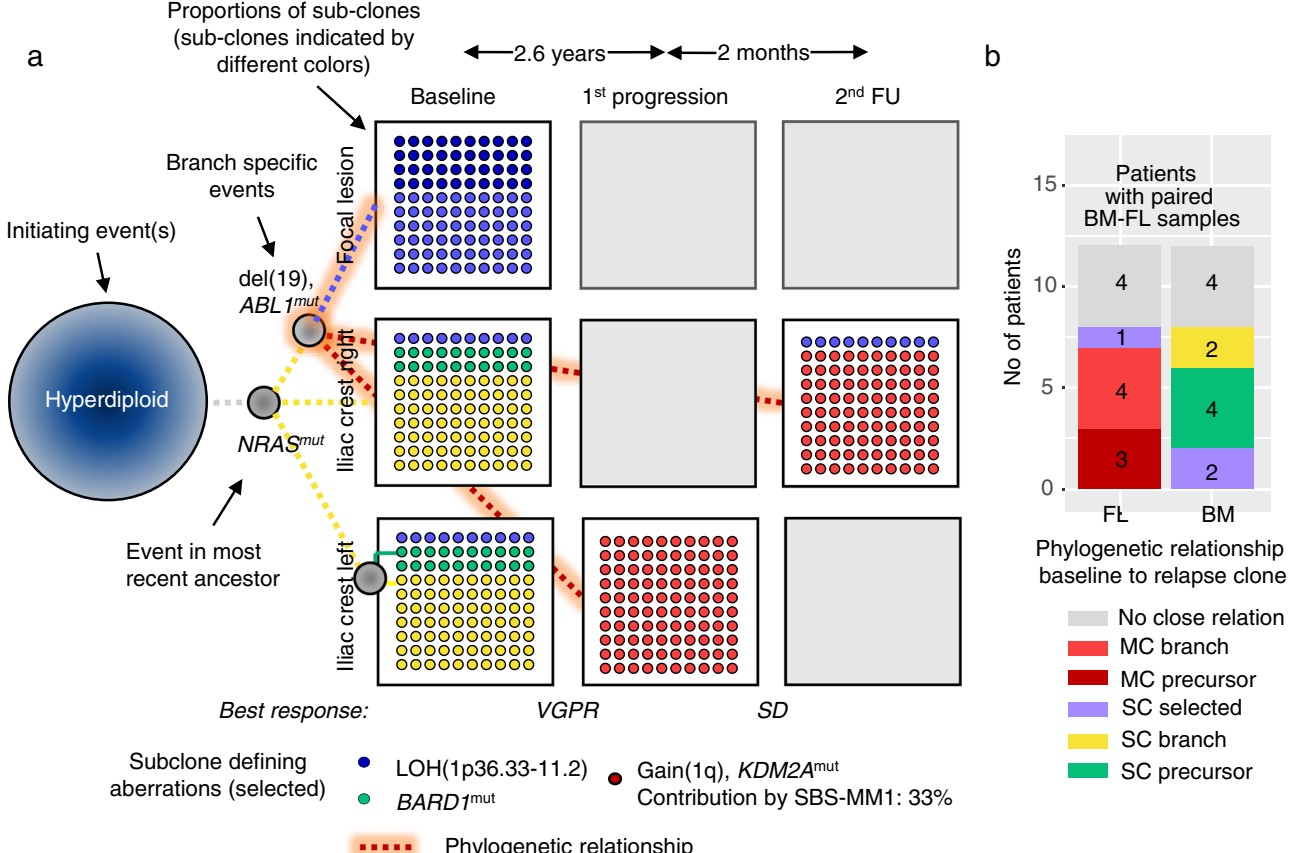

**Fig. 3 | Phylogenetic relationship between baseline and relapse clones.** We analyzed the phylogenetic relationship between baseline clones from different bone marrow regions and relapse clones to elucidate the origin of resistant disease. To highlight the unknowns in our sample/dataset, we have chosen a format with flipped and unflipped (blank) cards reminiscent of a memory game. In **a** the mock phylogenetic tree for P6 is shown as an example for a pattern where the unique baseline focal lesion clone and the dominant relapse clone share the phylogenetic branch. The boxes show the proportion of detected subclones, with each subclone having a distinct color. For instance, ten blue dots indicate a proportion of 10% for the "blue" subclone. Dashed lines illustrate the origin/relationship of these sub-clones and the color code corresponds to the color of the subclones. The gray box/card indicates that tumor data is not available for the respective site at relapse.

A timeline of treatment and sampling alongside a detailed mock oncogenetic tree is shown in Suppl. Fig. 8. FU follow-up, VGPR very good partial remission, SD stable disease. **b** For 12 patients with paired baseline iliac crest/focal lesion samples, the relationship is shown separately for the focal lesion (FL, left bar), and the iliac crest (BM, right bar). No close relationship (gray): baseline and relapse clones represent different evolutionary branches as shown for the dominant iliac crest and the relapse clone in **a**; selected: a preexisting resistant clone is selected (example in Suppl. Fig. 1); branch: the two clones share the same branch, as shown for the baseline focal lesion and the relapse clone in **a**; precursor: linear relationship between baseline and relapse clone, i.e., the clone acquires new aberrations; SC subclone is related to relapse clone, MC major/dominant clone is related to relapse clone. Source data are provided as a Source Data file.

available sample per patient, collected after a median of 5 treatment lines (range: 1–15) and 4.9 (1.4–13.2) years. On average, 41% (0–82%) of mutations were major in one sample but not detectable in the other (Fig. 2d). The spatiotemporal changes affecting known myeloma drivers are shown in Fig. 2e. Despite these huge changes in clonal composition, baseline and relapse were clonally related, even in patients with more than 10 years of follow-up, with the first and last dominating subclone having the same most recent common ancestor in all cases. Considering just the number of sub-clones per patient and sample, these changes would not be apparent. We observed, on average, three subclones both at baseline and after therapy (Suppl. Fig. 1). However, taking all time points into consideration, the total number of subclones per patient was significantly higher with an average of 5 (range 2–11, $p < 0.01$). Mock oncogenetic trees for all patients are shown in Suppl. Figs. 2–27.

Taken together, this data suggests that significant spatial genomic heterogeneity is associated with focal lesions, while the clonal composition of the diffuse MM infiltrate in the pelvis tends to be more homogeneous both at baseline and at subsequent distinct time points posttreatment. Further, we show similar numbers of subclones at

baseline and after therapy but major BM-wide changes in subclonal genetic composition in longitudinal analyses.

## The relationship between focal lesion clones and resistant disease

A major implication of our initial findings is that preexisting clones derived from focal lesions contribute to changes in clonal composition observed posttreatment. To address the hypothesis, we analyzed longitudinal genetic data from patients, for whom paired iliac crest/focal lesion specimens at baseline and BM samples posttreatment were available. In the majority of these cases ($n = 7/12$), we observed a close relationship between the baseline clonal composition of the focal lesion and the relapse clone(s), with the major focal lesion subclone either being the precursor or sharing a phylogenetic branch with the relapse clone(s) (Fig. 3a and Suppl. Figs. 8, 10, 12, 15, 17, 22, 25). Except for one of these patients, the clonal branch leading to relapse was also detectable at the iliac crest but in five of them solely at a minor subclonal level, supporting the results of a recent baseline-relapse study[18] (Fig. 3b).

Taken together, the precursors of resistant disease can frequently be found as dominant subclones in focal lesions, suggesting focal lesions as a major site of advanced disease.

### Single clone expansion vs. multiple surviving subclones

Despite the close phylogenetic relationship between focal lesion clones and relapsed resistant disease states, we did not detect the unique relapse clone as a major subclone in the baseline focal lesion sample with only 2/12 patients with paired baseline iliac crest/focal lesion specimens showing the selection of a preexisting minor subclone at the iliac crest (Fig. 3b and Suppl. Figs. 9, 10). One possible explanation for this lack of a tight association is that relapse is driven by a very small unique subclone or even a single cell in the majority of patients, as has recently been shown in the "single-cell expansion" model[16], making its detection at baseline highly challenging.

To test this hypothesis, we analyzed 21 patients for whom baseline and at least two follow-up samples were available, and performed a longitudinal analysis of the clonal architecture. All patients received melphalan (Suppl. Data 1), which, with some limitations (see methods), allowed us to use the mutational signature SBS-MM1 (melphalan) as a barcode for single-cell tracking. In one-third (7/21) of patients, all detectable MM cells in the follow-up samples originated from one expanded subclone at relapse, and in five of them, the signature SBS-MM1 was detectable in mutations, which were common to all relapse cells (Fig. 4a, Suppl. Figs. 3, 19–124, and Suppl. Tables 1, 2). Since the underlying mutations are only detectable through bulk sequencing when one single cell expands, these observations further support the existence of a single-cell expansion mechanism. However, in two-thirds ($n = 14$) of patients, we found evidence that more than one subclone (typically two) from the primary untreated tumor was involved in the seeding of relapse cells. In ten of them, multiple subclones were seen at the same location at least once during follow-up, and we also noted changes in subclonal proportions in eight of these examples, reminiscent of the clonal competition model[15] (Fig. 4b and Suppl. Figs. 4, 7–11, 14, 16–18, 26).

In the remaining four patients with multi-clone resistant disease, we identified a special pattern, where unique subclones from different branches are seen at distinct locations, in contrast to the classical clonal competition model, where multiple genetically distinct subclones coexist at the same location. This "alternating spatial clonal dominance" pattern is illustrated in Fig. 5 and Suppl. Fig. 27. A timeline of treatment and sampling for each of these patients alongside detailed mock oncogenetic trees is shown in Suppl. Figs. 12, 13, 15, 25. Taken together, we observed three patterns of subclonal evolution in MM that occur over time in response to cycles of remission and relapse, which are (1) single-cell expansions and sweeps, (2) coexisting expanding (and competing) subclones, and (3) site-unique expansions of distinct subclones with the main difference between the second and the third pattern being the anatomical location of subclones.

### The link between observed evolutionary patterns and clinical features

We investigated whether the three evolutionary patterns, which we observed, were associated with specific clinical features. Due to the limited sample size, we restricted the analysis to the risk status defined by the GEP70 and the number of focal lesions on PET-CT at baseline, which are independent prognostic factors[12,19]. Furthermore, we included best response during first-line therapy as another parameter, as this was previously shown to be associated with branching evolution in MM[5]. Risk status at baseline was not associated with the evolutionary patterns seen at relapse. In contrast, achieving CR was a significant determinant of the type of MM evolution. All patients in the single-cell expansion and alternating spatial clonal dominance categories had relapsed from CR but more than half of the patients with coexisting subclones (i.e., in the clonal competition model) had only achieved PR ($n = 3$) or VGPR ($n = 3$) as their best response ($p < 0.05$, fisher's exact test, Fig. 6). A suboptimal treatment response could be linked to the intensity of treatment. Indeed, we found deviations from the total

therapy approach at UAMS in 9/21 patients, including single or no autologous stem cell transplantation and/or less than a triplet combination during maintenance. This was seen in 7/10 patients with competing treatment-resistant subclones and 2/7 patients with a single detectable expanding subclone. While this difference was not significant, we cannot exclude a link between evolutionary patterns and treatment intensity during first-line therapy.

A strong association with the evolutionary patterns was seen for the number of focal lesions detected at baseline (Fig. 6). In all patients with ≥3 PET-positive focal lesions, an imaging pattern associated with poor outcome[12,14], relapse was driven by multiple surviving subclones, whereas in patients with <3 PET-positive focal lesions an expansion of MM cells originating from a single precursor was typically seen at the observed time points(7/10 patients, $p < 0.01$, fisher's exact test). Of note, the majority of total follow-up samples ($n = 7/9$), which were collected from patients with the spatial dominance pattern, were from focal lesions as the concomitant diffuse myeloma infiltrates at the iliac crest was low or even absent, and MM cells could not be enriched for WES. This macrofocal relapse pattern was recently associated with poor outcome[20] and indeed, three of the patients in our set died within 4 years (patients #10, #11, #23). Both, lack of paired iliac crest samples as well as early deaths could explain the lower number of follow-up samples in this group as compared to the other two evolutionary patterns.

In summary, these observations suggest that the single-cell expansion model is more relevant for patients with none or only a limited number of PET-positive focal lesions at baseline. The data also indicate that a higher number of PET-positive focal lesions at baseline is associated with greater levels of spatial clonal heterogeneity, which increase the likelihood of multiple subclones being-able to survive treatment and lead to relapse.

### Mixture of evolution patterns

The three patterns of evolution, which we describe herein, were not mutually exclusive, with their components occurring either sequentially or in parallel in the majority of patients. On the one hand, all except one patient from the "single-cell (clone) expansion" group showed further clonal evolution and had evidence of subclonal competition during the course of the disease (Fig. 4a). On the other hand, 11 of the 14 patients with multiple surviving subclones showed the SBS-MM1 melphalan exposure signature in at least one subclone (Suppl. Table 2). Intriguingly, we observed a single-cell expansion in a focal lesion at the right sacrum and concomitant selection of a preexisting clone without evidence of SBS-MM1 at the right iliac crest in the patient #9 (Suppl. Fig. 28).

The best example of a mixed pattern of clonal evolution was seen in patient #12 (Fig. 7 and Suppl. Fig. 14). This patient had a homogeneous clonal composition at the left and right iliac crest at baseline. During their follow-up over 14 years, three resistant clones were detected, which had the same preexisting precursor but which were comprised of one of three unique clonal branches driven by three independent single-cell expansions. While the first clone dominated in focal lesions, the second was seen primarily at the iliac crest. The third clone, defined by a del(17p) among others, emerged 13 years after diagnosis following salvage ASCT and was not seen in any of the prior samples, suggesting that it had been in a dormant state for years. This clone dominated at the iliac crest at the last time point.

### The frequency of clonal sweeps

Selective clonal sweeps, which refer to an expansion of a new clone and suppression of previously dominant clones, are a crucial component of MM evolution[2,3,15]. Sweeps are clinically relevant since the respective subclones could have new drug susceptibilities and resistance profiles. However, the frequency as well as the clinical context in which clonal sweeps occur remain poorly understood. To get deeper insights into

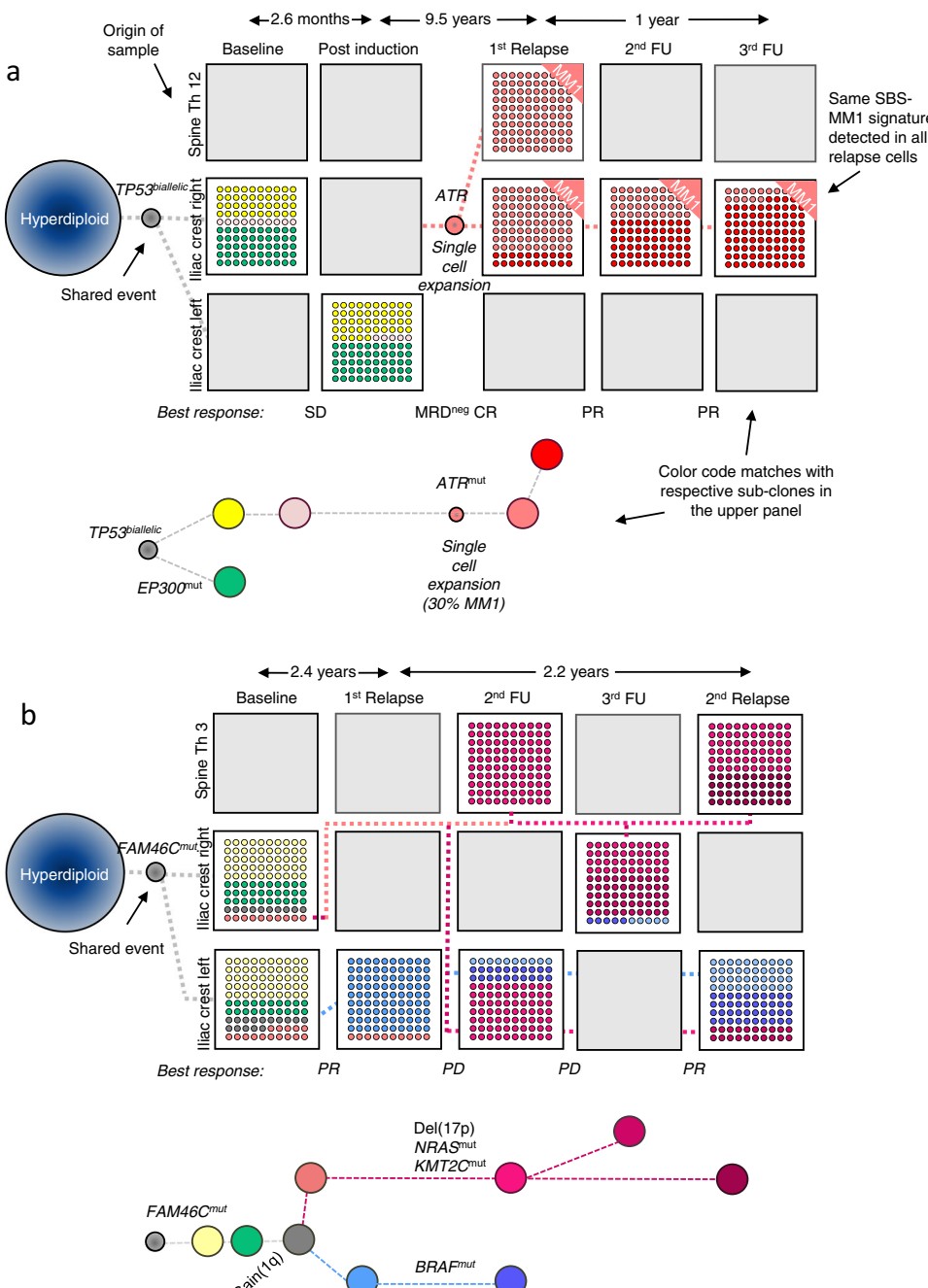

**Fig. 4 | Evolutionary patterns during treatment.** In **a** the mock phylogenetic tree for patient #22 is shown as an example of a pattern where relapse is initially driven by a single cell. Subsequently, the first relapse clone is replaced by a further evolved descendant subclone. Expansion of a single cell was demonstrated by a mutational signature analysis which showed a significant contribution of the melphalan signature SBS-MM1 to relapse clone-defining mutations. A timeline of treatment and sampling alongside a detailed mock oncogenetic tree is shown in Suppl. Fig. 24. In **b** two independent branches drive relapse and longitudinally alternate in dominance in patient #14 which resembles the classical alternating clonal dominance model by ref. 15. Yet, clonal dominance also alternates in space. Due to space constraints data for another left iliac crest sample after the sixth treatment line (second FU in the plot) is not presented but shown in detail in Suppl. Fig. 16. The boxes show the proportion of detected subclones, with each subclone having a distinct color. Dashed lines illustrate the origin/relationship of subclones/branches. For instance, in **b** all subclones belonging to the "red branch" are interconnected by a red dashed line. Selected clone-defining mutations are shown for each branch, with known myeloma drivers being in bold. The length of branches does not indicate the extent of differences between clones but the time point of their first appearance. Gray boxes/cards indicate that tumor data is not available for the respective time points and sites. FU follow-up, PR partial remission, SD stable disease, PD progressive disease, MRDneg CR minimal-residual-disease negative complete remission. Source data are provided as a Source Data file.

this process, we analyzed 14 patients with at least two follow-up iliac crest specimens and observed no, 1, 2, and 3 sweeps in 1, 5, 6, and 2 patients, respectively, during the observation period. Clonal sweeps were not seen exclusively in patients relapsing from CR, but rather were seen in relapse from all response levels, including the presence of major clonal changes in patients with progressive disease (Suppl. Table 3). We assume that a high proliferation rate and accelerated seeding of a more resistant subclone[3,16] masked the extinction of previously dominant subclones in these patients. With each clonal sweep, the dominant subclone(s) became more and more advanced, as defined by the acquisition of novel mutations and/or CNAs. Thus, clonal sweeps are a common evolutionary event in MM during which

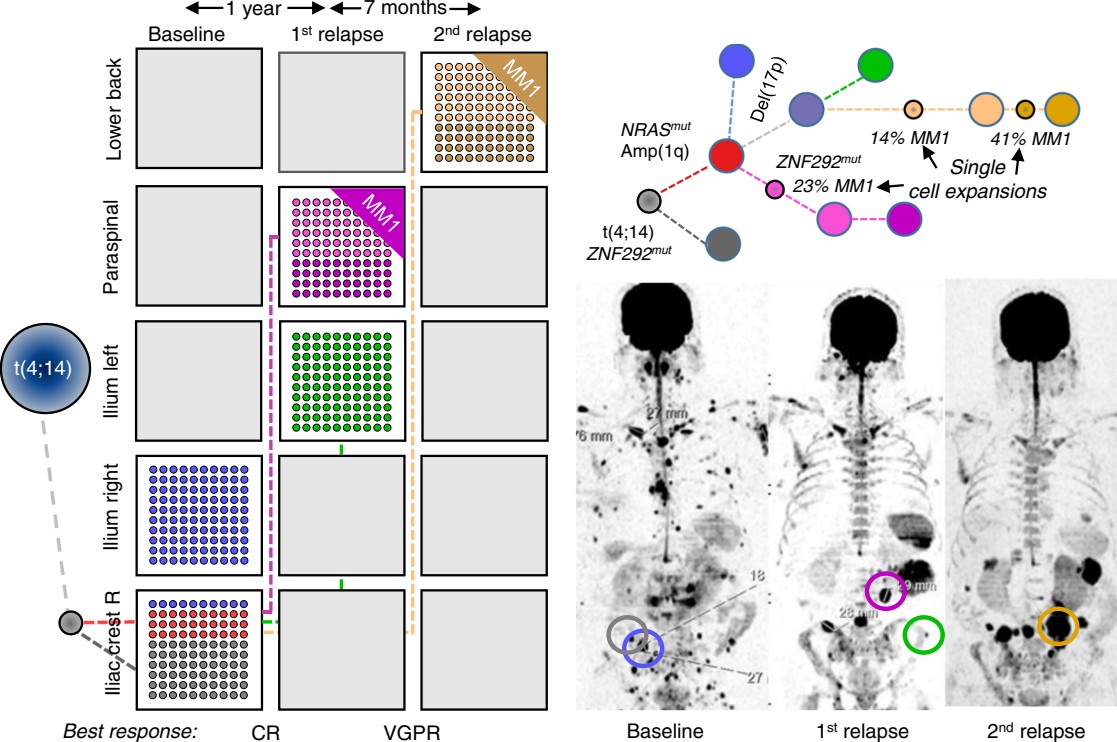

**Fig. 5 | Alternating spatial clonal dominance pattern.** MRI images for patient #23 at baseline and at two follow-up time points, as well as the corresponding mock phylogenetic tree. In this patient, unique subclones were seen at distinct locations after relapse. The boxes show the proportion of detected subclones, with each subclone having a distinct color. Dashed lines illustrate the origin/relationship of subclones/branches. For instance, the subclones belonging to the "pink" and "green" branch are interconnected by a pink and green dashed line, respectively. Known myeloma drivers are marked in bold. The tree is aligned with the functional imaging plots and as such, the length of branches does not indicate the extent of subclonal differences. A timeline of treatment and sampling alongside a detailed mock oncogenetic tree is shown in Suppl. Fig. 25. VGPR very good partial remission, CR complete remission. Source data are provided as a Source Data file.

the evolutionary fittest clones come to dominate in the BM cavity, raising the question as to the underlying drivers of fitness.

### Spatiotemporal parallel evolution

The identification of parallel evolution by the analysis of spatial heterogeneity can provide a tool to identify the key drivers of relapse[21]. Parallel evolution refers to a pattern in which subclones evolve through independent mutations, targeting the same genes or the same molecular pathways[21]. Focusing on *NRAS*-, *KRAS*-, and/or *TP53*-mutated subclones, we observed this type of evolution in five patients in our data. Patient #13 presented with two unique del(17p)/*TP53*^mut double-hit events at different locations at baseline and another *TP53* double-hit at relapse (Suppl. Fig. 27). Subclones with almost identical driver mutations were seen in patient #24, where the first branch, which dominated at baseline, had concomitant mutations in *NRAS*, *DIS3*, and del(17p), while the second branch, which emerged at relapse, had mutations in *KRAS*, *TP53*, and *DIS3* (Suppl. Fig. 26). Both branches were associated with single-cell expansions, highlighting the commonality between the subclones. Stable evolution of two coexisting *KRAS*-mutated subclones was observed in patient #2 at the iliac crest (Suppl. Fig. 4). Yet, this patient developed an interval focal lesion at the left ischium that was related to one of the *KRAS*-branches but showed a site-unique *IKBKB* mutation and del(2-q37.2-q37.3), illustrating evolutionary changes in a patient with seemingly stable disease.

We extended the analysis to examine other genes recently implicated as MM drivers[22], and strikingly, we observed multiple mutations in epigenetic modifiers of the lysine demethylase (KDM)/methyltransferase (KMT) family in patient #12, who presented with three clonal branches, *KDM4B* and *KMT2D* were each affected in a branch (Fig. 7 and Suppl. Fig. 14). In total, we found mutations in these two classes of epigenetic modifiers in 10/25 patients (Suppl. Table 4). While one patient showed *KDM* mutations at baseline, in all ten patients, mutations in *KDM* and/or *KMT* genes emerged during treatment. Since they appeared in resistant diseases and evolved in parallel, mutations in the *KDM/KMT* gene family are promising candidates as relapse drivers in MM, further supporting the role of epigenetic modifiers in the evolution of high-risk disease[23].

Together, our analysis demonstrates spatially separated parallel evolution and supports mutations in epigenetic modifiers as promising candidates for drivers of MM relapse.

## Discussion

In this longitudinal multi-region sequencing study of MM, we provide a detailed picture of MM evolutionary pathways developing during therapy. Patients included in the study were extensively treated in Total Therapy trials or similar protocols and were salvaged with multi-agent therapies, including a range of novel drugs available at that time. Due to the long follow-up time, we could include all molecular subgroups and even patients with long-lasting remissions of up to 10 years during frontline therapy.

The distinct genetic makeup of focal lesions[11], which was confirmed in this study, and their prognostic impact are consistent with them having a significant role in MM progression. In this study, we show a close relationship between baseline focal lesions and relapse clones. Yet, it was challenging to track down the site and nature of the unique preexisting relapse clone at disease presentation, which could be due to the relatively low number of samples and the limited sensitivity of WES. However, we think that the mutagenic impact of treatment and recent advances in the analysis of mutational signatures provide better explanations. MM cells usually acquire new mutations when exposed to alkylating agents[24] and in our set, ~1/3 of new

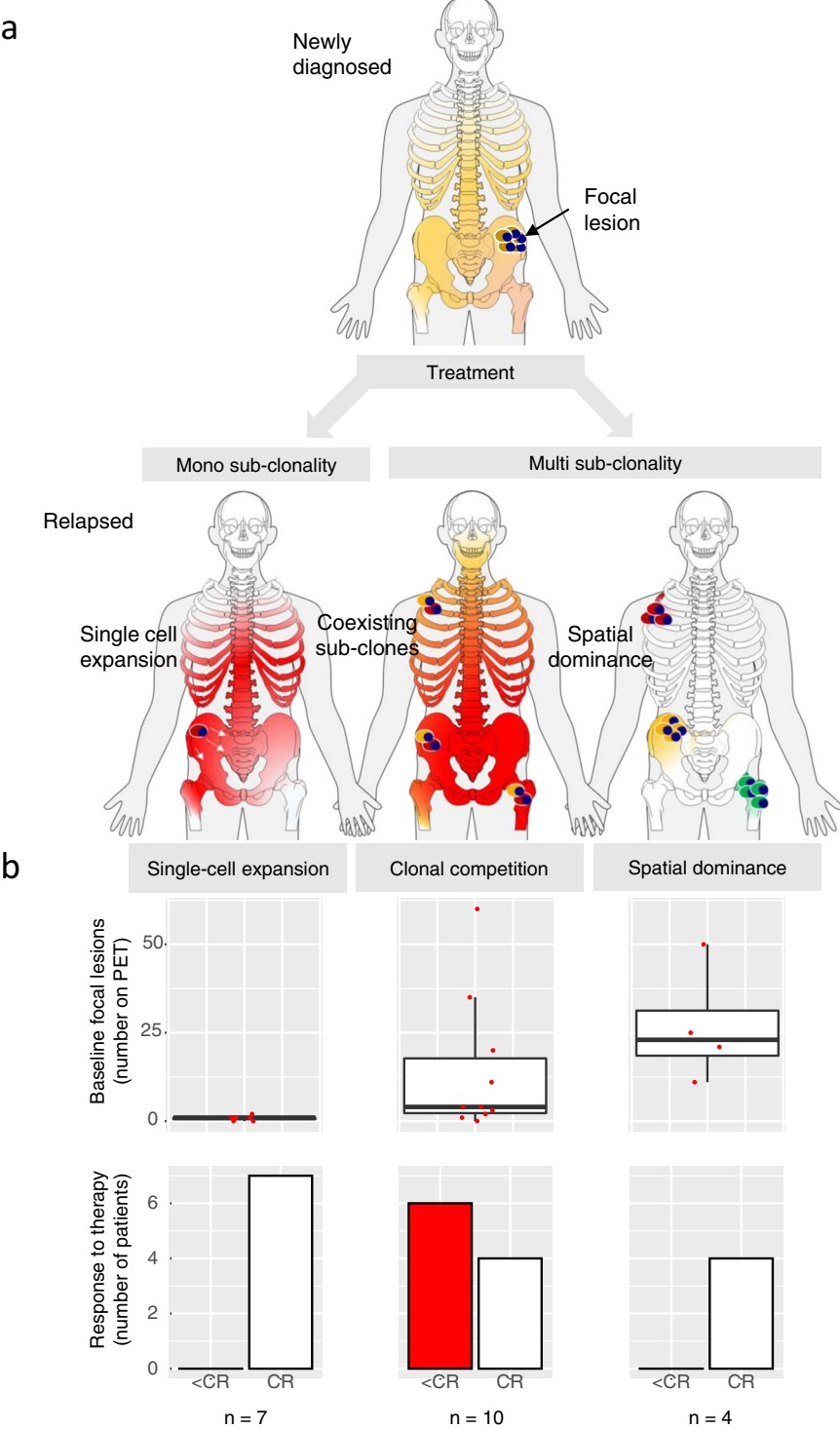

**Fig. 6 | Evolutionary patterns and association with clinical features.** In **a** the three evolutionary patterns, which we observed in this study, are illustrated. These include (1) expansion and sweep driven by single tumor cells, (2) coexisting expanding subclones, and (3) site-unique expansions of distinct subclones, with the main difference between the second and the third pattern being the anatomical location of subclones. In **b**, boxplots are shown for the association between these patterns and (1) the number of PET-positive focal lesions at baseline (upper panel) and (2) the response to first-line therapy (lower panel), respectively. The boxplots show the median and the interquartile range, while the upper and lower whiskers show the highest and lowest values (excluding outliers), respectively. CR complete remission. Source data are provided as a Source Data file.

mutations could be linked to melphalan. As a result, clonal changes are generally inevitable during exposure to multi-agent chemotherapy. Second, and even more important, single-cell expansions were often seen at relapse, in line with recent predictions based on samples from postmortem cases[16]. As these single cells could be from anywhere in the BM, even the most sensitive technologies would not be able to detect them, especially if spatial heterogeneity is not considered.

Given the nature of the MM clonal architecture, which can include site-specific subclones, it is difficult to capture the full extent of clonal diversity in MM. It seems that MM evolution is driven by a limited

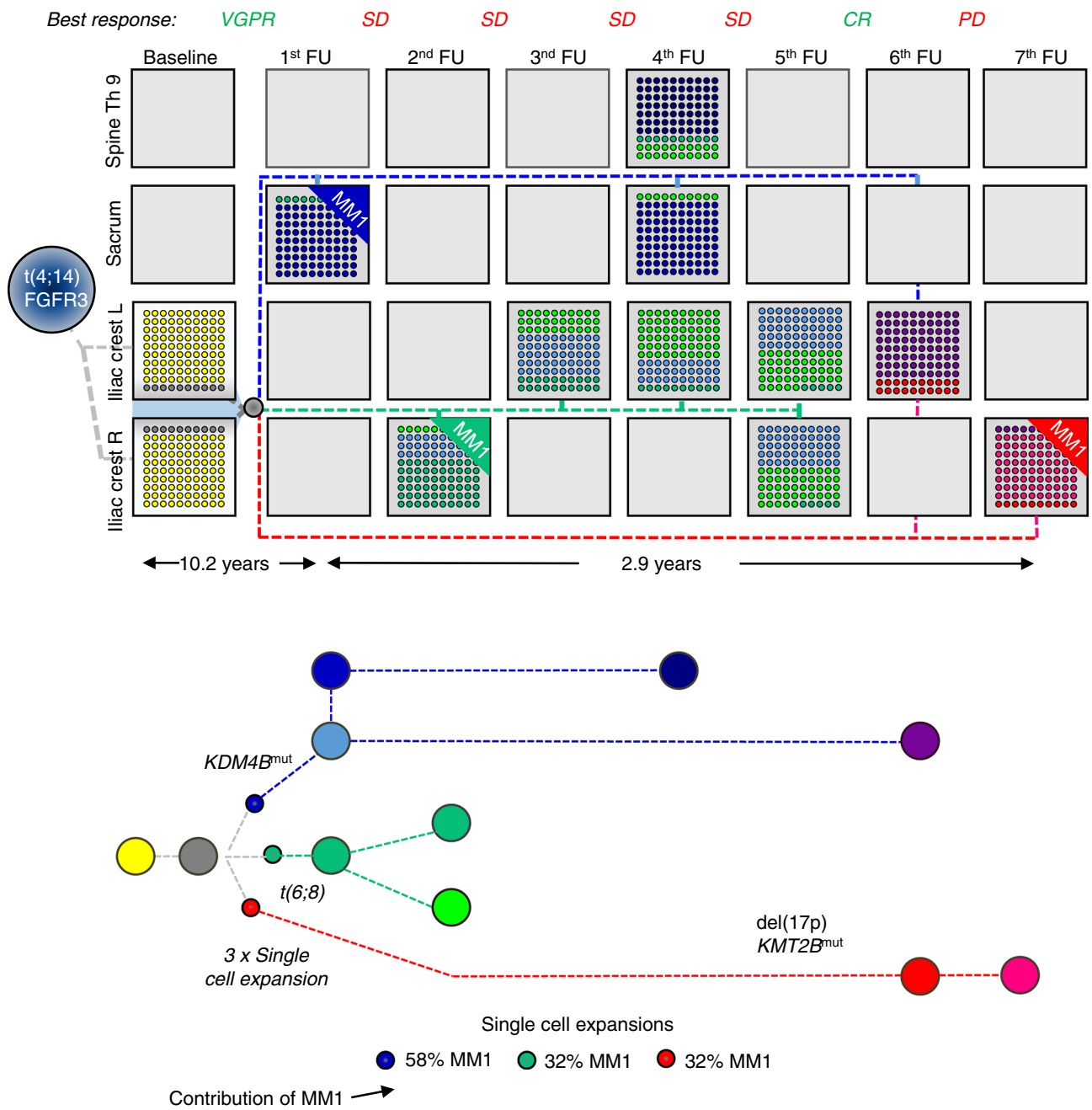

**Fig. 7 | Spatial-longitudinal alternating dominance of single-cell expansions.** At baseline, patient #12 presented with a t(4;14) and the same clonal composition was seen at the left and right iliac crest. The first follow-up (FU) sample was collected from a sacrum lesion after the sixth treatment line. The dominant subclone (blue branch) was positive for the SBS-MM1 signature, indicating a single-cell expansion. The precursor (gray) of this subclone was already detectable at baseline. Two months later, a sample was collected from the right iliac crest. While the focal lesion subclone was detectable at the minor level, another SBS-MM1-positive subclone with an MYC translocation (green branch) dominated at this location, suggesting clonal co-existence or competition. Despite extensive treatment, a very similar clonal composition was seen in two focal lesions and six iliac crest samples. However, after another autologous stem cell transplantation (sixth FU), the patient achieved a complete remission (CR) and a subclone emerged, which was not seen in prior samples (red branch). This del(17p) and signature SBS-MM1 positive subclone,

which dominated at the iliac crest at the last available time point, shared the pre-existing precursor with the other two relapse subclones but represented another branch. Single-cell expansions were predicted based on the presence of the SBS-MM1 melphalan signature. The boxes show the proportion of detected subclones, with each subclone having a distinct color. Dashed lines illustrate the origin/relationship of subclones/branches. For instance, all subclones of the "blue branch" are interconnected by a blue dashed line. The length of branches of the phylogenetic tree corresponds to the time point of the first appearance of respective clones. For convenience, only known myeloma drivers and mutations affecting two members of the *KDM/KMT* gene family are shown in the mock phylogenetic tree. A more detailed description is shown in Suppl. Fig. 14. Please note that due to limited space, data for a sample after the 11th treatment line is only presented in the Suppl. Fig. 14. SD stable disease, PD progressive disease, VGPR very good partial remission. Source data are provided as a Source Data file.

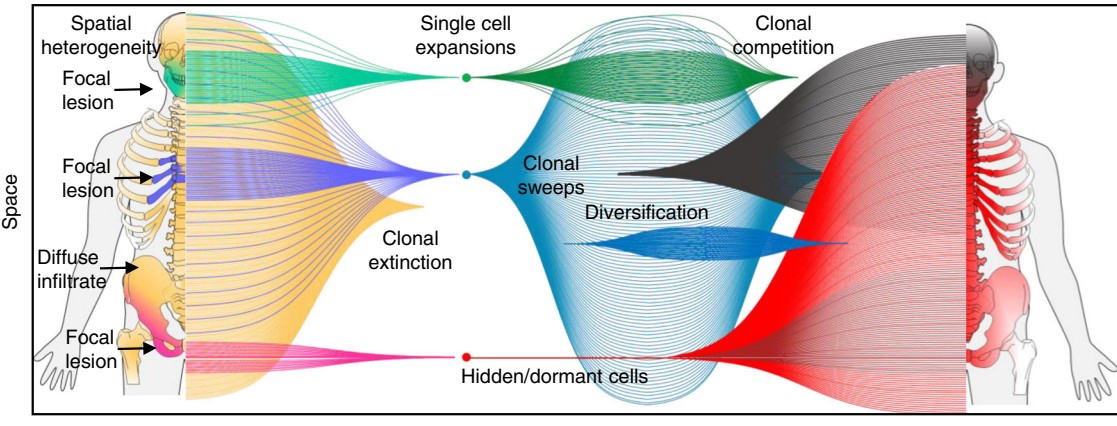

**Fig. 8 | Summary of the spatial-temporal evolution of multiple myeloma.** The figure summarizes our findings and the complexity of myeloma evolution seen in this study. It includes spatial heterogeneity at baseline and after treatment, a link between subclones and growth patterns, resistant clones that may be hidden/ quiescent over many years of follow-up, as well as the emergence of multiple selective clonal sweeps derived from single cells, diversification of expanding subclones and competition between them.

number of major subclones that generate the majority of diversity at relapse. However, despite this speculation, we reported previously on a patient with unique subclones present in four different focal lesions[11]. Notably, this patient had >100 focal lesions, suggesting a tremendously higher amount of genomic heterogeneity than is seen in the current limited set of patients and samples, especially as myeloma cells from the iliac crest were not available for each time point and many focal lesions were not biopsied. The same subclonal diversity likely holds true for treated patients, where we observed the emergence of unique subclones after multiple relapses and >10 years of treatment, clearly supporting a clonal composition which is dominated by clones with the highest proliferation rate and potential for adaptation. As a result of this observation, it is highly likely that relapse is associated with hidden subclonal heterogeneity and that our view on evolution is still incomplete. This complex subclonal architecture is further supported by recent longitudinal studies, which showed the emergence of previously undetectable subclones in relapsed-refractory MM patients treated with IMIDs, BRAF inhibitors, or anti-BCMA CAR T cells[5,6,25,26].

From a therapeutic perspective, it is important to understand why some relapsed patients present with multiple detectable subclones, whereas in others, relapse is driven by just one single clonal expansion. Here we demonstrate that the depth of response is associated with the number of (detectable) resistant subclones. While single-cell expansions were only seen in patients achieving CR, all patients relapsing from less deep responses had multiple detectable surviving subclones. A further important variable we describe is that the likelihood of multiple subclones surviving treatments and leading to progression is significantly increased in patients with multiple PET-positive focal lesions at baseline. This not only demonstrates the value of incorporating imaging data together with molecular analyses but also provides one biological explanation for the negative prognostic impact of focal lesion number and size[12,13,27]. It seems to be that multiple unique advanced subclones at different BM locations increase the chance of there being multiple resistant subclones that can mediate a short time to progression. Yet, we appreciate that our findings need to be validated in larger spatiotemporal studies.

Our observation further underscores the need for functional imaging at diagnosis and for response assessment[9,28,29]. For instance, in some patients, multiple surviving subclones did not coexist at the same location, as described by ref. 15, but dominated at different locations; a pattern which we called "alternating spatial clonal dominance". This pattern can also be seen in a subset of newly diagnosed patients[11]. We speculate that the potential of tumor cells to circulate via the peripheral blood is restricted in these patients but appreciate that the underlying mechanism remains elusive. As a variation of this unique pattern, we observed patients with spatial clonal dominance in which there was some exchange between sites.

Critically examining the sequencing data, we only considered mutations, which were seen in ≥20% of MM cells in at least one of the patient's samples in order to avoid an overestimation of heterogeneity. As a result, the study was focused on partially expanded subclones and thus rather underestimated the full extent of diversity. Hence, our approach could be enhanced in the future by using single-cell methods[30–32]. Yet, it is important to keep in mind that MM cells may be subject to Muller's ratchet, a phenomenon characterized by continuous accumulation of aberrations, including deleterious mutations[33]. However, as clinicians and cancer biologists, we have a major interest in the few mutations and ancestors that drive progression and treatment resistance[34].

In conclusion, the study highlights the complexity of MM evolution, including spatial heterogeneity during treatment, a link between subclones and growth patterns (focal vs. diffuse), resistant clones that may be hidden/dormant over many years of follow-up, as well as the emergence of multiple selective clonal sweeps derived from single cells (Fig. 8). We show the value of medical imaging as a tool to identify patterns of MM evolution, which allowed us to provide one possible explanation for the negative prognostic impact of baseline focal lesions. The massive clonal changes over time, even in relapsed MM patients with seemingly stable disease, will make it necessary to assess the current genomic profiles if targeted therapies are considered. Last but not least, the presence of single cells that can drive relapse even after 10 years of remission suggests that a prerequisite for curative therapies would be to overcome not only tumor heterogeneity but also dormancy.

## Methods
### Patients and samples
The study was approved by the institutional review board of the University of Arkansas for Medical Sciences (#02815), and all patients signed written consent in accordance with the Declaration of Helsinki. We performed whole-exome sequencing (WES) of CD138 purified biobanked MM cells, which had been collected from 24 patients at the University of Arkansas for Medical Sciences between 2003–2017. The patients were enrolled in total therapy protocols[35–37] or treated with a

total therapy-like frontline regimen including multi-agent induction therapy, stem cell transplantation, and intensified maintenance therapy (Suppl. Data 1). The patients did not receive compensation. For the comparison of paired samples in treated patients, we included specimens which were collected within 6 weeks of the same treatment line. We discriminated between randomly collected samples from the iliac crest and CT-guided specimens from focal lesions, representing diffuse tumor cell infiltration and nodular tumor cell accumulations, respectively. In total, our study included 140 tumor samples, including 92, 42, 4, and 2 iliac crest specimens, focal lesion biopsies, soft tissue specimens, and peripheral blood plasma cell leukemia samples, respectively. CT-guided biopsies were taken from all types of bone and extramedullary lesions, excluding lesions located in the long bones due to the risk of peri-interventional fracture (Suppl. Data 1). Not all patients and samples were included in each analysis (Suppl. Table 5). In order to validate our recent findings, patients which were included in our previous analysis of spatial heterogeneity[11] were not considered for comparisons of paired samples. Therefore, we excluded paired samples from patients #10, #13, #16, and #23 (baseline) as well as #9, #12, #22, and #24 (treated).

## Whole-exome sequencing and mutation calling

Tumor and control DNA were isolated from CD138-positive PCs and peripheral blood leukapheresis products collected after induction therapy, respectively. WES libraries were prepared using the SureSelectQXT sample prep kit in combination with the SureSelect Clinical Research Exome kit (Agilent) or the HyperPlus kit (Kapa Biosystems) combined with the SeqCap EZ MedExome kit (Roche Nimblegen). The exome kits contained additional baits covering the Ig and *MYC* loci. De-multiplexing of raw paired-end sequencing data was performed using bcl2fastq v2.20.0. Reads were aligned to the human genome reference GRCh37 release 75 (https://grch37.ensembl.org/index.html) using BWA-mem version 0.7.12[38]. Sambamba v0.5.6[39] was used to sort and index bam files and to mark duplicates. Somatic single-nucleotide variants (SNVs) were called using MuTect v1.1.7[40] and Strelka v2.9.10[41] with default parameters. The intersection of SNVs identified by both variant callers was filtered using the fpfilter.pl script (https://github.com/ckandoth/variant-filter) with default parameters. After exclusion of variants located in immunoglobulin loci, we determined read counts for all mutations and samples per patient using the Rsamtools R package v1.24.0 and the following inclusion criteria: unique reads, coverage exceeding 20× in all samples of the patient, a mapping quality of at least 20 and base quality of at least 20 at the site of the variant. To maintain a conservative approach and avoid an overestimation of heterogeneity, we only included SNVs with a cancer clonal fraction (CCF) of ≥0.20 in at least one sample (corresponding to a clonal proportion of 20%) and called this SNP in the paired sample(s) if at least two variant reads were detected (usually, MuTect only calls mutations if at least three variant reads are detected). Furthermore, for heterogeneous mutations, we performed manual somatic variant refinement using IGV v2.8.6[42] according to a published standard operating procedure[43]. This filtered set of SNVs, which had a median coverage of 121x, was annotated using SNPeff v4.3[44] and used for downstream analyses. Missense, nonsense, splice-site, and frameshift SNVs were defined as non-silent. For the analysis of spatiotemporal changes affecting known myeloma drivers, we only considered variants with a combined annotation-dependent depletion (CADD) score >20[45]. For the description of heterogeneity, we used the same terminology as described previously[11]. Briefly, we called mutations *shared*, if they were present in both samples with the same or similar CCF. We classified mutations with at least a threefold difference in CCF as shared-differential ("*shared-diff*"). We called mutations *unshared*, if they were detectable in only one of the paired samples and discriminated between minor(CCF < 60%) and major (CCF ≥ 60%) mutations.

## Translocation calling and copy number profiling

Ig and MYC translocations were identified using Manta v1.5.0[46]. Translocations with somatic variant quality scores <30 or "imprecise" calls were removed. All translocation calls were manually inspected in IGV[42]. For annotation of translocations, we used ANNOVAR v2017.07.01[47]. CNAs were called using Sequenza v2.1.2. For each sample, the accuracy of copy number calls was verified by manual inspection of LogR and BAF values for each CNA. To avoid overcalling heterogeneity we used a threshold of 5 Mb for global CNA analyses as described[11]. For the detection of deletions affecting MM driver genes, we used a threshold of 1 Mb.

## Evolutionary patterns and subclonal reconstruction

The CCF was calculated as described[48]. Briefly, the mutation copy number was determined using the following equation:

$$n_{\text{mut}} = f_s \times \frac{1}{p} \left[ p n_{\text{locus}}^t + 2(1-p) \right] \tag{1}$$

where $n_{\text{mut}}$ is the mutation copy number, $f_s$ is the fraction of mutated reads (variant allele frequency), $p$ is the tumor purity, and $n_{\text{locus}}^t$ is the locus-specific copy number. For $p$ and $n_{\text{locus}}^t$ we used the values predicted by Sequenza. We then compared the expected $f_s$ value to values assuming the mutation was on 1, 2, 3, …, C chromosomes and assigned $n_{\text{chr}}$ the value of C with the maximum likelihood using a binomial distribution. Finally, the CCF was determined by dividing $n_{\text{mut}}$ by $n_{\text{chr}}$. Clonal substructures were inferred using SciClone v1.1.0[49] with the filtered set of SNVs and default parameters, except for minimum depth, which was set to 50. For the manual design of mock phylogenetic trees, the output of SciClone was further interpreted after the inclusion of copy number data. Subclones were defined based on SciClone clusters and the presence of at least two mutations or at least one copy number aberration. To identify expansions of single tumor cells during treatment we applied a recently published strategy based on mutational footprints of chemotherapies[16,50,51]. This strategy is based on the fact that the underlying mutations are only detectable through bulk sequencing when single cells expand. Briefly, we first fitted subclone-defining mutations in each patient (cut-off of 25 mutations, median: 57, 410 range: 25–405) with the mutational signature single-base substitution (SBS)-MM1 (melphalan exposure) and the latest COSMIC reference (https://cancer.sanger.ac.uk/cosmic/signatures/SBS/) for SBS1, SBS2, SBS5, SBS8, SBS9, SBS13, and SBS18 using *mmsig* v02.02.2020 (https://github.com/evenrus/mmsig) according to the authors' recommendations[6,17,52]. The presence of SBS-MM1 was further confirmed using the mSigAct v0.9 signature presence test (https://genome.cshlp.org/content/suppl/2018/04/09/gr.230219.117.DC1) and a *p* value cut-off of 0.05 to account for the limited number of mutations in WES[50,53].

## Derivation of the GEP70 risk signature

For risk stratification, we applied the GEP70 model, which is based on Affymetrix U133Plus2.0 microarray data (Santa Clara, CA) for CD138-enriched PCs[19]. Raw intensity values were MAS5 normalized, converted to log2 scale, and corrected for batch effects using M-ComBat[54]. The GEP70 corresponds to the average log2 expression of 51 upregulated genes minus the average log2 expression of 19 downregulated genes, and scores ≥0.66 indicate high risk.

## Medical imaging

PET with CT attenuation correction (PET-CT) and diffusion-weighted magnetic resonance imaging with background suppression (DWIBS) were done as recently described[13]. Briefly, PET-CT was performed on a Biograph 6 PET/CT system (Siemens Medical Solutions, PA, USA), a GE Discovery IQ scanner (GE Healthcare, IL, USA) or a CTI-Reveal scanner (Siemens Medical Solutions). Images were acquired from the vertex to the toes. After iterative reconstruction, images were reviewed using

the PET volume computer-assisted reading software (AW server, version 3.2, General Electric, WI, USA). DWIBS was performed on a 1.5 Tesla Achieva scanner (PHILIPS, MA, USA). Scanning was performed from vertex to toes in 7 to 9 slabs, depending on the patient's height. A coronal whole-body T1 turbo spin echo image was used as a localizer. Images were analyzed in an inverted grayscale with fused whole-body maximum-intensity projection reconstructions of the diffusion and exponential apparent-diffusion coefficient images. For PET-CT, a focal lesion was defined as a circumscribed focus with increased FDG uptake compared to its surroundings. For DWIBS, a focal lesion was defined as a well-delineated focal intensity above the surrounding background BM ≥1 cm in size.

### Statistical methods

Statistical analyses were carried out using the R software package 3.6.0. Group comparisons of continuous variables were done using the Mann–Whitney–Wilcoxon test for independent groups. Differences in treatment responses and evolution patterns between groups were assessed using Fisher's exact test.

### Reporting summary

Further information on research design is available in the Nature Research Reporting Summary linked to this article.

## Data availability

The raw WES dataset has been deposited in the dbGAP database under accession number phs2625.v1. Due to individual privacy concerns, the data were available under restricted access. Access may be requested by permanent employees of their institution at a level equivalent to a tenure-track professor or senior scientist with responsibilities such as laboratory administration and oversight. The requests are managed by the Data Access Committee of the NCI, and after approval, access is permitted for 12 months. The remaining data were available within the Article, Supplementary Information, or Source Data file. Source data are provided with this paper.

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

## Acknowledgements

This work was supported by P01 CA 55819 from the National Cancer Institute. L.R. was funded by the Deutsche Krebshilfe via the MSNZ program Würzburg, and the Interdisciplinary Center for Clinical Research Würzburg (IZKF). N.W. was supported by the National Institute of General Medical Sciences of the National Institutes of Health under Award Number P20GM125503. A.M.P. and N.W. were supported by the Dietmar-Hopp Foundation. F.M. was supported by the American Society of Hematology and by Sylvester Comprehensive Cancer Center NCI Core Grant (P30 CA 240139). G.J.M. was supported by the Leukemia and Lymphoma Society.

## Author contributions

Conception and design: L.R. and N.W., Provision of study material or patients: G.J.M., B.B., F.v.R., M.Z., S.T., C.S., and F.E.D., Sample management and processing: S.D., Bioinformatic and statistical analyses: C.A., M.A.B., A.M.P., and N.W., Additional analyses: L.R. and C.S., Data interpretation: L.R., C.S., F.M., B.A.W., O.L., G.J.M., F.v.R., and N.W., Wrote the paper: L.R., C.S., F.M., O.L., G.J.M., and N.W., Reviewed and approved the paper: All authors.

## Funding

## Competing interests

The authors declare no competing interests.
