## [Peer review file · Nature Communications]

REVIEWER COMMENTS

Reviewer #1 (Remarks to the Author): Expert in multiple myeloma genomics and therapy

In “The Spatio-Temporal Evolution of Multiple Myeloma from Baseline to Relapse-Refractory States”, the authors apply whole exome sequencing to image-guided and longitudinally acquired samples to characterize spatial and temporal heterogeneity in myeloma patients. These data in turn serve to define the evolutionary patterns that characterize development of relapse/refractory disease. The study nicely complements and adds to the analysis previously published by the authors that reported on spatial heterogeneity in FLs vs. BM iliac crest samples. The current study was well designed, and the dataset, in particular, is highly novel, unique, and important as a resource for the myeloma genomics community. However, the analyses are a bit superficial in some parts. The authors show extensive genetic heterogeneity and manually reconstruct phylogenetic patterns to reflect the evolutionary path to relapse. However, their claims of 3 distinct evolutionary patterns is not validated and thus are somewhat speculative.

Specific comments and suggestions:

Clarification on the dataset. The authors indicate in the methods that not all samples were included into each analysis as some were included in a previous analysis. I presume that these are samples 10, 13, 16 and 23 that were not included in Figure 2c analysis. I think this should be more clearly indicated in the paper. Similarly data for treated Figure 2c is missing for patients 12 (2 timepoints) and 24. Although the regimens each patient received is indicated in the supplemental tables, it would be relevant to know what drugs the patient were receiving at the time of progression. It would be helpful to know if patients were progressing on a triplet vs one agent or a doublet (often patients stop some of the drugs because of toxicity for example) this is important information when considering pattern of progression where one might presume to see more clonal competition model for patients progressing on a single or even double agents as seen in the publication by Keats. Also the authors indicate that some samples were taken two months apart. I have concerns with inclusion of these samples as with 2 months of treatment could significantly alter the subclonal composition as has been shown by Keats et al and it is possible that a focal lesion biopsied after two months of treatment from when the bone marrow sampling could be enriched for resistant cells and that would be driving the differences between the two sites. Either these patients should be removed or at least indicate which samples were taken within two months but after treatment was initiated.

On page 4, the authors state that the largest level of genomic heterogeneity was seen when comparing the first to the last available sample per patient (Figure 2d). However, the explanation of what comparisons were included is not clear. More specifically, what was compared to conclude that the first and last samples are most different? Were all temporal samples compared pairwise? Additionally, what

does largest level of genomic heterogeneity mean? It maybe helpful to include in the legend the time between sample collection and number of lines of treatment.

Figure 3A demonstrates the data for patient 6. Figure 1 demonstrates collection of bone marrow at iliac crest at progression (NOT paired -indicated in black circle) yet this figure shows the subclones in both right and left iliac crest at first progression. From this analysis, they conclude that there is a close phylogenetic relationship between baseline FL and relapse subclones, however unless I am not understanding these results there is also a relationship between baseline BM and relapse BM based on this figure (figure 3b same number of focal and BM with no close relationship). Therefore the last paragraph of Page 4 is overstated in implying that the focal lesion is responsible for the relapse clone. This is speculative without clear evidence to demonstrate this.

Figure 4a. Again figure one indicates only one iliac crest for baseline sample yet both right and left iliac crest are shown. It does not appear that a samples was drawn at second PD yet subclones are shown at that timepoint. Please explain. Similarly for 4b it does not appear that a bilateral aspirate was taken at timepoint 3rdPD. Also is the FL tested at 2nd PD and 4th PD from the same site as other data (Figure S28) would indicate that FL from different sites would be more heterogeneous then what is shown here.

Based on their analysis the authors interestingly propose three distinct patterns of sub-clonal evolution in response to therapy and during cycles of remission and relapse. However, evidence for “Spatial Dominance” in Pt 13 (Supp Figure 28) is not sufficient. Can you show in additional patient that demonstrated this model (Pt. 10,11,23?). Also, in the abstract you say that this model is characterized “unique treatment-resistant subclones at distinct BM locations”. Shouldn’t it be “distinct locations”, not “distinct BM locations”. Pt 13 only has baseline BM sample, rest are focal lessons so cannot comment on if treatment resistance is also observed in the BM.

Importantly, the authors demonstrate that the evolution patterns are associated with distinct clinical features which make biological sense based on response however is there also an influence of therapeutic pressure as was previously demonstrated by Keats et al. As mentioned would be informative to have included data on drugs that patients were immediately refractory to at the time of sampling.

Overall the data are intriguing and supportive of the conclusions however numbers are small, many patients do not have all corresponding samples, many focal lesions are not biopsied and timing of match samples is questionable so data is somewhat speculative.

Minor comments

Would be helpful if patient illustrated in the figures were explicitly provided in the text (for example, in the second last paragraph of Page 4, second paragraph on Page 5 – so readers can make the connection to each patient in Supplementary Figure 2-27).

For consistency can Supplementary Figure 28 be included as Figure 4c.

The authors can be somewhat vague about reporting results in the text. For example, the first paragraph on Page 5, the authors report “only a few patients showing selection ...”. Consider explicitly stating the absolute numbers.

Can the authors clarify in the Figure legends what the coloured dashed lines represent in Figures 3,4, and 6.

No Table Legend on Supplementary Table 4. Needs explanation -don't know how to interpret.

Reviewer #2 (Remarks to the Author): Expert in computational cancer genomics and cancer evolution

The study “The Spatio-Temporal Evolution of Multiple Myeloma from Baseline to Relapse- Refractory States” details the genomic evolution of multiple myeloma. To this end the authors have generated whole exome sequencing data from 144 specimens sampled from 25 MM patients over the course of up to 14 years. These data are being used to reconstruct the clonal evolution of the disease in each patient. The authors report distinct evolutionary patterns at relapse by single or multiple subclones that exhibit varied patterns of spatial segregation. A pattern of multi-clonal relapse was associated with a greater number of focal bone lesions. The authors find evidence that relapse can be seeded by subclones that went undetected over decades.

A number of studies, in part from some of the co-authors of this study, have investigated the evolution of MM using genomic sequencing. These studies have revealed complex patterns of clonal evolution over the years of disease progression and regression. Such study also demonstrated that relapses can be monoclonal (Landau, 2020) as well as multiclonal (Bolli, 2014).

I can't see a major flaw of the study, but I'm afraid I also fail to see how the study adds more than incremental insight over existing publications or if there are specific highlights their presentation feels underdeveloped.

One drawback is that the ability of categorising mono and multi-clonality can be limited by the number of samples taken, so I'd caution against drawing too strong conclusions. It should be made clear how sampling and time relate to the number of detected clones and the proposed patterns of evolution. As is the classification of single cell expansion / clonal competition / spatial dominance appears somewhat artificial. The association of the evolutionary pattern with therapy outcomes could be the most interesting part, but it requires more careful analysis / presentation to be truly convincing.

In that light the presentation of the paper could be improved. It would be very helpful to have (i) a timeline of treatment and sampling for each patient, alongside (ii) evolutionary trees that contain full genomic annotation, including driver gene mutations and the branch lengths.

The latter would be important to shed light on the nature of emerging dormant clones at relapse - do they exhibit shorter branch lengths than previously existing clones and potentially also different signatures? (Possibly difficult from WES data)

What exactly is the take home from Figure 7, other than 'it's complicated'?

How were driver gene mutations defined - any coding change in a gene related to multiple myelomagenesis?

Did the authors call indels? The methods only refer to SNVs.

It would be helpful if the authors clearly stated in the abstract/introduction that their analysis is based on whole exome sequencing.

The skeleton is larger than the human ideograms used.

Reviewer 1

- 1) In “The Spatio-Temporal Evolution of Multiple Myeloma from Baseline to Relapse-Refractory States”, the authors apply whole exome sequencing to image-guided and longitudinally acquired samples to characterize spatial and temporal heterogeneity in myeloma patients. These data in turn serve to define the evolutionary patterns that characterize development of relapse/refractory disease. The study nicely complements and adds to the analysis previously published by the authors that reported on spatial heterogeneity in FLs vs. BM iliac crest samples. The current study was well designed, and the dataset, in particular, is highly novel, unique, and important as a resource for the myeloma genomics community. However, the analyses are a bit superficial in some parts. The authors show extensive genetic heterogeneity and manually reconstruct phylogenetic patterns to reflect the evolutionary path to relapse. However, their claims of 3 distinct evolutionary patterns is not validated and thus are somewhat speculative.

Response: First of all, we would like to thank the reviewer for the positive feedback that our study is well designed, highly novel, unique and important. We also appreciate the critique on the suggested evolutionary patterns and their validation. Our initial analysis was guided by the two established evolutionary patterns (Landau *et al.*, *Nature Communications* 2020 and Keats *et al.*, *Blood* 2012) and using our dataset and mutational signature analyses, we could confirm the presence of these two patterns, which are apparently in conflict with each other. One strength of our study is that we could show these clonal dynamics in both the spatial and temporal dimension. Furthermore, the spatial data allowed us to observe a special type of multi-clone resistant disease, which differed from the classical clonal competition model. In our opinion, the presence of unique sub-clones at distinct locations is important information for both diagnostics as well as molecular studies. Thus, we decided to introduce a third pattern. Yet, we appreciate that sampling could impact the interpretation of evolutionary patterns and we also highlight that there are mixtures of evolutionary patterns, suggesting that they are not necessarily mutually exclusive/distinct. To highlight the unknowns in our sample/data set, we have chosen a figure format with flipped and unflipped (blank) cards reminiscent of a memory game. Furthermore, we have performed additional analyses and discussed the limitations of our study throughout the manuscript more carefully. All changes in the text are marked in yellow. Please see also the reply to your concerns #3, and #12 and reviewer 2 critiques #2 and #3.

- 2) The authors indicate in the methods that not all samples were included into each analysis as some were included in a previous analysis. I presume that these are samples 10, 13, 16 and 23 that were not included in Figure 2c analysis. I think this should be more clearly indicated in the paper. Similarly, data for treated Figure 2c is missing for patients 12 (2 timepoints) and 24.

Response: We agree with the reviewer that the exclusion of samples was not well described in the first version of the manuscript. One aim of the comparison of paired samples was to validate the findings of our previous manuscript on spatial genomic heterogeneity using an independent set of patients. Thus, we excluded paired samples from patients #10, #13, #16, and #23 (baseline) as well as #9, #12, #22 and #24 (treated), which were presented in our previous study (Rasche *et al.*, *Nature Communications* 2017). To address the reviewer’s comment, we revised the method section: “In order to validate our recent findings, patients which were included in our previous analysis of spatial heterogeneity¹¹ were not considered

for comparisons of paired samples. Therefore, we excluded paired samples from patients #10, #13, #16, and #23 (baseline) as well as #9, #12, #22 and #24 (treated).” Furthermore, we revised the legend of Figure 2: “For this comparison, paired samples from patients #10, #13, #16, and #23 (baseline) as well as #9, #12, #22 and #24 (treated) were not considered, as they had already been included in our previous analysis of spatial heterogeneity¹¹.”

3) *Although the regimens each patient received is indicated in the supplemental tables, it would be relevant to know what drugs the patient were receiving at the time of progression. It would be helpful to know if patients were progressing on a triplet vs one agent or a doublet (often patients stop some of the drugs because of toxicity for example) this is important information when considering pattern of progression where one might presume to see more clonal competition model for patients progressing on a single or even double agents as seen in the publication by Keats.*

Response: In our opinion the reviewer raises an important question regarding the impact of doublets versus triplet combination on evolution patterns. To address this comment, we have studied all patient records in detail and looked for treatment reductions. We have updated Suppl. Table 2 accordingly and, as requested by Reviewer #2, we show a timeline of treatment and sampling for each patient alongside evolutionary trees in Suppl. Fig. 3-26. Focussing on first-line therapy, we indeed observed differences between the patterns, which are not significant but support the assumption of the reviewer. We found deviations from the standard treatment approach at UAMS in 9/21 patients, including just single or no autologous stem cell transplantation and/or less than a triplet combination during maintenance. This was seen in 7/10 patients with competing treatment-resistant subclones but just 2/7 patients with a single expanding subclone. We describe this finding in the revised manuscript: A suboptimal treatment response could be linked to the intensity of treatment. Indeed, we found from the total therapy approach at UAMS in 9/21 patients, including single or no autologous stem cell transplantation and/or less than a triplet combination during maintenance. This seen in 7/10 patients with competing treatment-resistant subclones and 2/7 patients with a single expanding subclone. While this difference was not significant, we cannot exclude a link between evolutionary patterns and treatment intensity during first-line therapy.

4) *Also the authors indicate that some samples were taken two months apart. I have concerns with inclusion of these samples as with 2 months of treatment could significant alter the subclonal composition as has been shown by Keats et al and it is possible that a focal lesion biopsied after two months of treatment from when the bone marrow sampling could be enriched for resistant cells and that would be driving the differences between the two sites. Either these patients should be removed or at least indicate which samples were taken within two months but after treatment was initiated.*

Response: We would like to thank the reviewer for pointing to this issue. As already outlined in the response to point #3, we went through all patient records. In the revised version of the manuscript, we now present sampling time points and treatment lines in great detail in the updated Suppl. Fig. 3-26. The figures show that iliac crest-focal lesion pairs were collected within 1-2 week except for one patient. In the latter, paired samples were collected 6 weeks apart from each other, but the patient had a drug holiday during this time and thus was not treated as shown in Suppl. Fig. 21.

Yet, when checking the patient paper record for patient #25, we realized that another treatment line had already been initiated several weeks before collection of the aspirate from the left iliac crest. Although we did not see any major changes in the clonal composition, we did not consider the paired iliac crest samples for the comparison to make sure that all samples fulfilled our

inclusion criteria. Since samples of this patient had only been considered for the comparison of paired samples in the first version of the manuscript, we excluded the patient from our study and updated the manuscript accordingly.

5) *On page 4, the authors state that the largest level of genomic heterogeneity was seen when comparing the first to the last available sample per patient (Figure 2d). However, the explanation of what comparisons were included is not clear. More specifically, what was compared to conclude that the first and last samples are most different? Were all temporal samples compared pairwise? Additionally, what does largest level of genomic heterogeneity mean? It maybe helpful to include in the legend the time between sample collection and number of lines of treatment.*

Response: Our aim was to show that the average level of clonal heterogeneity between longitudinally collected samples can even exceed the average level of spatial heterogeneity observed in paired samples, which were collected at the same time point. We agree that our wording was misleading. Therefore, we revised the paragraph: “Compared to these paired samples, even larger differences in genomic profiles were seen, when we compared the first to the last available sample per patient, collected after a median of 5 treatment lines (range: 1-15) and 4.9 (1.4-13.2) years.” As requested by the reviewer, we added 1) time in years and 2) treatment line to Figure 2d.

6) *Figure 3A demonstrates the data for patient 6. Figure 1 demonstrates collection of bone marrow at iliac crest at progression (NOT paired -indicated in black circle) yet this figure shows the subclones in both right and left iliac crest at first progression.*

Response: We would like to thank the reviewer for pointing to this mistake. Indeed, the sample from the right iliac crest was collected after two months of second-line treatment with KRD with stable disease as best response. We have corrected the figure accordingly.

7) *From this analysis, they conclude that there is a close phylogenetic relationship between baseline FL and relapse subclones, however unless I am not understanding these results there is also a relationship between baseline BM and relapse BM based on this figure (figure 3b same number of focal and BM with no close relationship). Therefore the last paragraph of Page 4 is overstated in implying that the focal lesion is responsible for the relapse clone. This is speculative without clear evidence to demonstrate this.*

Response: We agree with the reviewer that our data do not prove that focal lesions are the origin of relapse disease. This is why we clarified in the manuscript that “we did not detect the unique relapse clone as a major sub-clone in the baseline focal lesion sample” and that “these single cells could be from anywhere in the BM”. Yet, relapse disease frequently had a close phylogenetic relationship to the major sub-clones in focal lesions. While these sub-clones were also often seen at the iliac crest, we usually only detected them at a minor sub-clonal level at this site, suggesting that focal lesions can be the major sites of these advanced sub-clones.

To address the reviewer’s comment, we revised the abstract (“While we did not find the unique relapse sub-clone as a major sub-clone in the baseline focal lesion(s), we show a close phylogenetic relationship between baseline FL and relapse disease, suggesting FLs as a major site of advanced disease.”) and the respective summary in the Results section: “Taken together, the precursors of resistant disease can frequently be found as dominant sub-clones in focal lesions, suggesting focal lesions as a major site of advanced disease.”

8) *Figure 4a. Again figure one indicates only one iliac crest for baseline sample yet both right and left iliac crest are shown. It does not appear that a samples was drawn at second PD yet subclones are shown at that timepoint. Please explain.*

Response: The second iliac crest sample was collected after induction to which the patient responded with stable disease. In the initial version, we showed both iliac crest samples on top of each other to demonstrate the similarity between these two samples, which were collected prior to first relapse. However, we absolutely agree with the reviewer that this could be misinterpreted and revised the figure accordingly. We also agree that the terminology for the time points was not precise. The patient did not respond to the 2nd treatment line, then was switched to KRd as 3rd line treatment and achieved a partial remission. The next follow-up sample was collected during the 3rd treatment line, while the last sample was collected when KRd was stopped due to toxicity. Thus, to be more precise, we now call these two time points second and third follow-up instead of progressive disease.

9) *Similarly for 4b it does not appear that a bilateral aspirate was taken at timepoint 3rdPD. Also is the FL tested at 2nd PD and 4th PD from the same site as other data (Figure S28) would indicate that FL from different sites would be more heterogeneous then what is shown here.*

Response: We have double checked the clinical annotation for patient #14, who is shown in Fig. 4b. The right iliac crest sample was collected only 25 days after the sample from the left site. Yet, we agree with the reviewer that the type of presentation was imprecise, since treatment was changed twice between these two samples. Furthermore, the sample from the left side was collected during the 6th line only 21 days after another sample from the same site during the 6th line and both samples showed a very similar clonal composition. For these reasons and also to account for the limited figure size, we decided to completely remove the second left iliac crest sample from this main figure. However, complete data is presented in Suppl. Fig. 16. In this patient, the focal lesion samples were taken from the same site. Indeed, the subclonal composition at that site remained relatively stable (descendant sub-clone appeared), while in patient #13 unique sub-clones were detected in focal lesions at different sites.

10) *Based on their analysis the authors interestingly propose three distinct patterns of sub-clonal evolution in response to therapy and during cycles of remission and relapse. However, evidence for “Spatial Dominance” in Pt 13 (Supp Figure 28) is not sufficient. Can you show in additional patient that demonstrated this model (Pt. 10,11,23?). Also, in the abstract you say that this model is characterized “unique treatment-resistant subclones at distinct BM locations”. Shouldn’t it be “distinct locations”, not “distinct BM locations”. Pt 13 only has baseline BM sample, rest are focal lessions so cannot comment on if treatment resistance is also observed in the BM.*

Response: To provide further evidence for the “Spatial Dominance” pattern, we now present data of patient #23 in the new Fig. 5 and data of patient #13 in Suppl Figure 28. In addition, detailed molecular and clinical data of all 4 patients are shown in the updated Suppl. Fig. 12, 13, 15 & 25. We agree with the reviewer that “distinct locations” is the correct expression since two of the patients had also extramedullary disease. In patient #13 the plasma cell infiltration at the

iliac crest was too low for a successful enrichment of CD138-positive myeloma cells for whole-exome sequencing or even absent at the follow-up time points, which is why we cannot present data for this location. We have revised the manuscript accordingly: Of note, the

majority of total follow-up samples (n=7/9), which were collected from patients with the spatial dominance pattern, were from focal lesions as the concomitant diffuse myeloma infiltrate at the iliac crest was usually low or even absent (data not shown).

- 11) *Importantly, the authors demonstrate that the evolution patterns are associated with distinct clinical features which make biological sense based on response however is there also an influence of therapeutic pressure as was previously demonstrated by Keats et al. As mentioned would be informative to have included data on drugs that patients were immediately refractory to at the time of sampling.*

Response: As requested, we have added this information to the Suppl. Figures 3-26. Please also see our response to point #3.

- 12) *Overall the data are intriguing and supportive of the conclusions however numbers are small, many patients do not have all corresponding samples, many focal lesions are not biopsied and timing of match samples is questionable so data is somewhat speculative.*

Response: We thank the reviewer for the overall positive evaluation! To take the reviewer's critique on board, we have rephrased the discussion and described further limitations of our study: "Given the nature of the MM clonal architecture, which can include site-specific sub-clones, it is difficult to capture the full extent of clonal diversity in MM. It seems that MM evolution is driven by a limited number of major sub-clones that generate the majority of diversity at relapse. However, despite this speculation we reported previously on a patient with unique sub-clones present in four different focal lesions¹¹. Notably, this patient had >100 focal lesions, suggesting a tremendously higher amount of genomic heterogeneity than is seen in the current limited set of patients and samples, especially as myeloma cells from the iliac crest were not available for each time point and many focal lesions were not biopsied. The same sub-clonal diversity likely holds true for treated patients, where we observed the emergence of unique sub-clones after multiple relapses and >10 years of treatment, clearly supporting a clonal composition which is dominated by clones with the highest proliferation rate and potential for adaptation. As a result of this observation, it is highly likely that relapse is associated with hidden sub-clonal heterogeneity and that our view on evolution is still incomplete." We also wrote: "Yet, we appreciate that our findings need to be validated in larger spatio-temporal studies."

- 13) *Would be helpful if patient illustrated in the figures were explicitly provided in the text (for example, in the second last paragraph of Page 4, second paragraph on Page 5 – so readers can make the connection to each patient in Supplementary Figure 2-27).*

Response: We would like to thank the reviewer for this excellent suggestion, which we think significantly improved the link between the main manuscript and the comprehensive supplemental material! As requested, we have provided the links to the Suppl. Fig. in the Results section.

- 14) *For consistency can Supplementary Figure 28 be included as Figure 4c.*

Response: We agree with the reviewer that presenting a patient with the spatial dominance pattern in the main manuscript is important. Due to lack of space and to account for the critique by reviewer #1, we created the new Figure 5, which shows data for patient #23, who also presented with the spatial dominance pattern.

- 0) *The authors can be somewhat vague about reporting results in the text. For example, the first paragraph on Page 5, the authors report “only a few patients showing selection ...”. Consider explicitly stating the absolute numbers.*

Response: We now provide the actual number of patients etc in the revised version of the manuscript.

- 1) *Can the authors clarify in the Figure legends what the coloured dashed lines represent in Figures 3,4, and 6.*

Response: Due to the complex spatio-temporal subclonal architecture in these patients, we decided to highlight the phylogenetic relationship between the sub-clones. Here, dashed lines illustrate the origin/relationship of sub-clones/branches. For instance, the red and blue dashed lines in Fig 4b indicate the two main branches to which the relapse sub-clones belonged to, with the color code corresponding to the primary sub-clone colors in this plot. We have added the following sentence to Fig. 4: “ For instance, in (b) all sub-clones belonging to the “red branch” are interconnected by a red dashed line.”

- 2) *No Table Legend on Supplementary Table 4. Needs explanation -don't know how to interpret.*

Response: We would like to apologize for this issue! We have added a title, which was missing in the previous version, and a legend to this table.

Reviewer 2

1. *I can't see a major flaw of the study, but I'm afraid I also fail to see how the study adds more than incremental insight over existing publications or if there are specific highlights their presentation feels underdeveloped.*

Response: First of all, we would like to thank the reviewer for the careful evaluation of our manuscript and the many helpful comments! Our study builds on recent longitudinal and spatial analyses of myeloma genomes. We confirm the presence of spatial heterogeneity in myeloma and using spatio-longitudinal data we show that there are indeed patients with a single-cell expansion mechanism. We agree with the reviewer that these are rather incremental insights.

Yet, we think that our study provides several novel insights, which are important to understand the spatio-temporal evolution of myeloma. We show for the first time:

- a rather homogenous sub-clonal architecture for the diffuse infiltration of myeloma.
- a close relationship between focal lesion sub-clones at baseline and relapse disease, suggesting focal lesions as a major site of advanced disease.
- a link between the number of PET-positive focal lesions at baseline and the number of sub-clones driving relapse, which provides one possible explanation for the negative prognostic impact of baseline focal lesions.

- evolutionary changes in a patient with seemingly stable disease, demonstrating the value of medical imaging as a tool to identify patterns of myeloma evolution.
- spatially separated parallel evolution
- mutated epigenetic modifiers as promising candidates for drivers of myeloma relapse

We have summarized these findings in the discussion. Yet, we agree with the reviewer that due to the multitude of observations, a novel key message might be difficult to be drawn from our study.

2. *One drawback is that the ability of categorizing mono and multi-clonality can be limited by the number of samples taken, so I'd caution against drawing too strong conclusions. It should be made clear how sampling and time relate to the number of detected clones and the proposed patterns of evolution. As is the classification of single cell expansion / clonal competition / spatial dominance appears somewhat artificial. The association of the evolutionary pattern with therapy outcomes could be the most interesting part, but it requires more careful analysis / presentation to be truly convincing.*

Response: We would like to thank the reviewer for this important comment. We also appreciate the proposed terminology “mono-” and “multi clonality”, which we now use in the revised figure 6. All changes in the text are marked in yellow.

- *Impact of sampling:* Sampling could indeed be one major confounder of categorizing mono- and multi-clonality, especially as “myeloma cells from the iliac crest were not available for each time point and many focal lesions were not biopsied.” To exclude that the described patterns were just due to differences in sampling, we compared the total number of treatment lines during the observation period and the total number of follow-up samples between these patterns. We did not detect differences between patients that were assigned to the single-cell expansion or the clonal competition model. However, as shown below, the numbers were lower for patients, who were assigned to the spatial dominance pattern:

At first glance, this seems to suggest a sampling issue. Yet, 3 of these patients met the criteria of macrofocal disease, which is defined by the presence of focal lesions and the absence of significant intervening bone marrow (BM) infiltration. Furthermore, 3 patients suffered from early death, which is in line with our (unpublished) observation of poor outcome for patients with a macrofocal pattern of disease relapse

(<https://doi.org/10.1182/blood.V128.22.4431.4431>). Both, early death and the lack of paired iliac crest samples, explain the lower number of total analyzed samples in this group. We have added this finding to the manuscript:

Of note, the majority of total follow-up samples (n=7/9), which were collected from patients in the spatial dominance category, were from focal lesions as the concomitant diffuse myeloma infiltrate at the iliac crest was usually low or even absent (data not shown). This macrofocal relapse pattern was recently associated with poor outcome (Ref Rasche) and indeed 3 of the patients in our set died within 4 years (patients #10, #11, #23). Both, lack of paired iliac crest samples as well as early deaths could explain the lower number of follow-up samples in this group as compared to the other two evolutionary patterns.

- *Artificial classification*: in this study we first discriminated between two published patterns of evolution: the single-cell expansion model by Maura and co-workers and the clonal competition model by Keats *et al.* Since these two models are apparently in conflict with each other and as a consequence the exact model of how clonal diversity is generated in myeloma during therapy remains controversial, we took advantage of our spatio-temporal data and addressed the question if and how often each of these evolutionary patterns can be observed in intensively treated myeloma patients. In one-third of patients all detectable MM cells in the follow-up samples originated from one single-cell/sub-clone from the primary tumor, and in most of these patients the signature SBS-MM1 was detectable in mutations, which were common to all relapse cells, indicating that the single-cell expansion pattern indeed exists in myeloma. The remaining patients showed more than one expanding sub-clone originating from the primary tumor at the observed time points. Yet, four patients had multiple sub-clones but at distinct locations and as such differed from the classical clonal competition model. Since this pattern could potentially impact both diagnostics as well as molecular studies, we decided to introduce a second category of multi-clonality, the spatial dominance pattern, to highlight that multi-clonality is not necessarily detectable at the iliac crest. Yet, we have edited Fig. 6 and show these two patterns now in the same category - "multi sub-clonality". Yet, we fully agree with the reviewer that even the classification of the two published patterns is somewhat artificial, since our data indicates a clonal composition which is dominated by clones with the highest proliferation rate and relapse seems to be associated with hidden sub-clonal heterogeneity.
 - *Conclusion*: Consequently, we appreciate the reviewer's comment that the results need to be interpreted with caution. Thus, we further edited the manuscript, e.g. we wrote that "we observed 3 patterns" instead of "there are 3 patterns", "further support" vs "confirm" and "the data indicate" instead of "the data is consistent with". We have also added to the discussion that "our view on evolution is still incomplete".
3. *In that light the presentation of the paper could be improved. It would be very helpful to have (i) a timeline of treatment and sampling for each patient alongside (ii) evolutionary trees that contain full genomic annotation, including driver gene mutations and the branch lengths. The latter would be important to shed light on the nature of emerging dormant clones at relapse - do they exhibit shorter branch lengths than*

previously existing clones and potentially also different signatures? (Possibly difficult from WES data)

Response: We would like to thank the reviewer for this suggestion, which in our opinion significantly improved the presentation of our data. As requested, we have added a timeline of treatment and sampling for each patient alongside evolutionary trees in the revised Supplemental Figures 3-26. Data for patient #12 is shown here as an example.

For each sample, we have added the time in days when the sample had been collected (e.g. B: baseline, 3742: days after initiation of treatment). The same information was added to Suppl. Table 1. In the plot, green arrows indicate treatment breaks and the length of arrows is proportional to the time interval. While we show the number of sub-clone defining events, the length of branches in the mock oncogenetic trees still does not indicate the extent of differences between clones because these were often a mix of somatic single-nucleotide and copy-number variants, with some of them being complex (e.g. acquired tetraploidy), making it difficult to calculate the extent of differences between sub-clones and as such the length of branches. For convenience we just present driver events in the oncogenetic trees of the main figures, since our major aim was to highlight the relationship between the relapse branches/sub-clones.

In our opinion it is a very interesting question, if novel expanding sub-clones exhibit shorter branch lengths than previously existing clones and show unique signatures. Regrettably, given the incomplete sample set and the nature of our data we can only partially address this

question. Due to the limited number of sub-clone defining mutations, our WES data was not suitable for such signature analyses. Emerging clones did not necessarily exhibit shorter branch lengths than previously existing clones in our analysis. On average, ~1/3 of new mutations could be linked to melphalan. Thus, in case a single-cell expansion of melphalan-exposed tumor cells was detected during later treatment lines, the respective sub-clone had relatively high numbers of sub-clone defining mutations. Patients #2 and #7, which are shown in detail in Suppl. Fig. 4 and 9, are examples for such a pattern.

4. *What exactly is the take home from Figure 7, other than "it's complicated"?*

Response: In our study we observed multiple components of myeloma evolution, including among others spatial heterogeneity before and during treatment, a link between sub-clones and growth patterns (focal vs. diffuse), resistant clones that may be hidden/dormant over many years of follow-up, as well as the emergence of multiple selective clonal sweeps derived from single cells. Due to the high numbers of components, we decided to visualize/summarize all these findings in an illustrative figure.

5. *How were driver gene mutations defined - any coding change in a gene related to multiple myelomagenesis?*

Response: In the first version of the manuscript we included all nonsilent (missense, nonsense, and splice site) single nucleotide variants in genes which had recently been identified as myeloma drivers by Walker and co-workers (Walker BA, *et al.* Identification of novel mutational drivers reveals oncogene dependencies in multiple myeloma. *Blood* 2018; **132**: 587–597.). Yet, to account for the reviewer's comment, we have calculated Combined Annotation Dependent Depletion (CADD) scores and considered only variants with a score >20 for the driver analysis in the revised version of the manuscript. As a result, *EP300* and *MAN2C1* were removed from Fig. 2.

6. *Did the authors call indels? The methods only refer to SNVs.*

Response: We focussed on single nucleotide variants and did not consider indels. In our experience, the variant allele frequency and the calculated cancer clonal fraction are not reliable for indels if whole exome sequencing is applied. As a result, major indels could potentially be classified as minor variants and vice versa, impacting the analysis of spatio-longitudinal changes in the sub-clonal composition.

7. *It would be helpful if the authors clearly stated in the abstract/introduction that their analysis is based on whole exome sequencing.*

Response: As requested, we state in the abstract and the introduction that our analysis is based on whole-exome sequencing.

8. *The skeleton is larger than the human ideograms used.*

Response: We thank the reviewer for carefully evaluating our manuscript. We wanted to highlight the skeletal system as the primary site of myeloma evolution, and therefore the skeleton is slightly larger than the ideograms.

REVIEWERS' COMMENTS

Reviewer #1 (Remarks to the Author):

The authors have very comprehensively addressed the original comments and made appropriate changes in the manuscript. I have no further comments.

Reviewer #2 (Remarks to the Author):

The authors have addressed my concerns and added helpful clarifications.

Revision of manuscript # NCOMMS-21-50104A

Editorial comments:

1. *Please see the author checklist.*

Response: We have replied to all comments/suggestions in the revised author checklist.

Reviewer 1

- 1) The authors have very comprehensively addressed the original comments and made appropriate changes in the manuscript. I have no further comments.

Response: We would like to thank the reviewer again for the positive feedback and the helpful comments!

Reviewer 2

- 1) The authors have addressed my concerns and added helpful clarifications.

Response: Thank you again for the constructive criticism!